# Comparative Immunogenicity of the Recombinant Receptor-Binding Domain of Protein S SARS-CoV-2 Obtained in Prokaryotic and Mammalian Expression Systems

**DOI:** 10.3390/vaccines10010096

**Published:** 2022-01-09

**Authors:** Iuliia A. Merkuleva, Dmitry N. Shcherbakov, Mariya B. Borgoyakova, Daniil V. Shanshin, Andrey P. Rudometov, Larisa I. Karpenko, Svetlana V. Belenkaya, Anastasiya A. Isaeva, Valentina S. Nesmeyanova, Elena I. Kazachinskaia, Ekaterina A. Volosnikova, Tatiana I. Esina, Anna V. Zaykovskaya, Oleg V. Pyankov, Sophia S. Borisevich, Arseniya A. Shelemba, Anton N. Chikaev, Alexander A. Ilyichev

**Affiliations:** 1State Research Center of Virology and Biotechnology “Vector”, Rospotrebnadzor, World-Class Genomic Research Center for Biological Safety and Technological Independence, Federal Scientific and Technical Program on the Development of Genetic Technologies, 630559 Novosibirsk, Russia; j.a.merkulyeva@gmail.com (I.A.M.); borgoyakova_mb@vector.nsc.ru (M.B.B.); shanshin_dv@vector.nsc.ru (D.V.S.); rudometov_ap@vector.nsc.ru (A.P.R.); lkarpenko@ngs.ru (L.I.K.); belenkaya.sveta@gmail.com (S.V.B.); isaevaanastasya93@gmail.com (A.A.I.); nesmeyanova_vs@vector.nsc.ru (V.S.N.); alenakaz@vector.nsc.ru (E.I.K.); volosnikova_ea@vector.nsc.ru (E.A.V.); esina_ti@vector.nsc.ru (T.I.E.); zaykovskaya_av@vector.nsc.ru (A.V.Z.); pyankov_ov@vector.nsc.ru (O.V.P.); ilyichev@vector.nsc.ru (A.A.I.); 2Laboratory of Chemical Physics, Ufa Institute of Chemistry, Ufa Federal Research Center, 450078 Ufa, Russia; monrel@yandex.ru; 3Federal Research Center of Fundamental and Translational Medicine, 630060 Novosibirsk, Russia; arseniya.shelemba@mail.ru; 4Institute of Molecular and Cellular Biology, Siberian Branch of the Russian Academy of Sciences, 630090 Novosibirsk, Russia; chikaev@mcb.nsc.ru

**Keywords:** SARS-CoV-2, COVID-19, subunit vaccines, S protein, receptor-binding domain

## Abstract

The receptor-binding domain (RBD) of the protein S SARS-CoV-2 is considered to be one of the appealing targets for developing a vaccine against COVID-19. The choice of an expression system is essential when developing subunit vaccines, as it ensures the effective synthesis of the correctly folded target protein, and maintains its antigenic and immunogenic properties. Here, we describe the production of a recombinant RBD protein using prokaryotic (pRBD) and mammalian (mRBD) expression systems, and compare the immunogenicity of prokaryotic and mammalian-expressed RBD using a BALB/c mice model. An analysis of the sera from mice immunized with both variants of the protein revealed that the mRBD expressed in CHO cells provides a significantly stronger humoral immune response compared with the RBD expressed in *E.coli* cells. A specific antibody titer of sera from mice immunized with mRBD was ten-fold higher than the sera from the mice that received pRBD in ELISA, and about 100-fold higher in a neutralization test. The data obtained suggests that mRBD is capable of inducing neutralizing antibodies against SARS-CoV-2.

## 1. Introduction

The COVID-19 pandemic caused by the SARS-CoV-2 virus has made vaccine development a top biomedical priority of modern healthcare. According to the WHO, at the beginning of 2021, more than 250 candidate vaccines against SARS-CoV-2 were at the clinical and preclinical study stages (https://www.who.int/publications/m/item/draft-landscape-of-covid-19-candidate-vaccines (accessed on 11 December 2021)). Various platforms have been used for creating vaccines based on protein S SARS-CoV-2 and its receptor-binding domain (RBD), including nucleic, vectored, and protein subunit vaccines [1]. Viral-vectored vaccines can elicit a specific immune response with neutralizing activity and protection, but they could also induce anti-vector immunity or present pre-existing immunity, causing some harmful immune responses. DNA and nanoparticle vaccines maintain a strong safety profile; however, they have lower immunogenicity. Subunit vaccines are generally safe without causing potential harmful immune responses, making them promising vaccine candidates. Moreover, subunit vaccines may target specific, well-defined neutralizing epitopes, with improved immunogenicity and/or efficacy [2,3,4]. In addition, subunit vaccines are easy to scale up to large-scale production; they have relative thermal stability and are suitable for shipment in a lyophilized form [5].

Currently, several subunit vaccines have been authorized for use around the world, including hepatitis B virus and Human papillomavirus vaccines that have already demonstrated their high efficiency [6,7]. One of the major drawbacks of subunit vaccines is a lower immunogenicity compared with attenuated vaccines. To enhance the potency of subunit vaccines, use of specific adjuvants and multiple injections are needed [8,9]. Another potential disadvantage of subunit vaccines is their relatively high manufacturing cost, especially if the mammalian expression system is used. Using prokaryotic vectors for protein expression could be more cost-effective; however, its antigenic specificity can be radically changed.

The aim of this study was to obtain a recombinant receptor-binding domain of protein S SARS-CoV-2 expressed in CHO and *E.coli* cells, and to compare the RBD’s immunogenic properties based on the test results of animals.

## 2. Materials and Methods

### 2.1. Creating the pET21-RBD, pVEAL2-RBD, pVEAL2-S, Expression Plasmids

The SARS-CoV-2 S gene (GenBank: MN908947) was codon-optimized and synthesized.

A DNA fragment encoding RBD domain of spike protein (a.a. 320V–542N) was amplified by PCR using SE-F (5′-aaaaaaggatccgtgcagcccaccgaatcc-3′) and SE-R (5′-aaaaaactcgaggttgaagttcacgcatttgttcttc-3′) primers, digested with BamHI and Sfr274I restriction enzymes, and cloned into a pET21 prokaryotic expression vector at the digestion sites to generate a pET21-RBD plasmid.

For expressing RBD in mammalian cells, integrative plasmid vector pVEAL2 was used. The RBD-coding fragment (a.a. 308V–542N) flanked by a DNA sequence-encoding tissue plasminogen activator (Tpa) signal peptide (MDAMKRGLCCVLLLCGAVFVSA) fused with His-tag was prepared by the overlap extension PCR method, using the following set of primers: Tpa-F (5′-gaccgccatgttggcattg-3′) and Tpa-R (5′-cagcagcacacagcagagccctctcttcattgcatccatggtggccccggggctagcctatagtgag-3′); TpaRBD-F (5′-tgctgtgtgctgctgctgtgtggagcagtcttcgtttcggccgtggaaaagggcatctaccagac-3′) and RBD4-R (5′-aaaaaagtcgacgaggctgatcagcggtttaaac-3′). The resulting PCR product was digested with Sfr274I and SalI, and cloned into a pVEAL2 expression vector.

In order to provide the expression of the trimeric spike (S) protein of SARS-CoV-2 (S-trimer) in the CHO cell line, DNA sequence encoding for the 1M-P1213 gene fragment of S protein was sub-cloned into the pVEAL2-S vector. Gene design was performed according to the publication of T. Li et al. [10]. Briefly, the protease cleavage-site-encoding fragment was removed from the S-protein gene, and mutations leading to K986P and V987P amino-acid-stabilizing substitutions were introduced. T4 bacteriophage fibritin trimerization domain and His-tag were inserted in the C-terminus of the protein. The new fragment was sub-cloned into the pVEAL2 vector.

### 2.2. Production of Recombinant Proteins

*E.coli* BL21 (DE3) cells were transformed by 100 ng of pET21-RBD plasmid. Transformed cells were grown at 37 °C in Luria–Bertani medium, and recombinant gene expression was induced with 1mM of isopropyl β-d-1-thiogalactopyranoside (IPTG) by standard protocol [11].

The S-trimer and mRBD proteins were stably expressed in a Chinese hamster ovary cell line (CHO-K1). Cells were transfected with the pVEAL2-S and pVEAL2-RBD plasmids, respectively, using Lipofectamine 3000 (Invitrogen, Carlsbad, CA, USA) in accordance with the manufacturer’s instructions. In order to integrate the vector expression cassette into the host genome, cells were co-transfected with the pCMV (CAT) T7-SB100 plasmid-encoding SB100 transposase. Transfected cells were selected with puromycin (10 µg/mL) for 3 days. Next, high-producing clones were isolated by dilution cloning, and cultured in roller bottles at 37 °C on DMEM/F-12 medium supplemented with 2% FBS.

### 2.3. Protein Purification

Pellets from IPTG-induced *E.coli* BL21 (DE3) cells were resuspended in a Lysis buffer (8M urea, 30 mM NaH_2_PO_4_, 20 mM imidazole, 0.5 mM NaCl, 1% Triton X-100 (*w*/*v*), pH 7.4) and disrupted by repeated sonication at 22 kHz; cell lysate was then centrifuged and supernatant was purified using a Ni-NTA column (Qiagen, Hilden, Germany). Eluted pRBD was then refolded via step dialysis against PBS with urea gradient (4 M, 2 M, 1 M, etc.).

Next, pRBD was additionally purified by ion-exchange chromatography, using SP-sepharose (cation exchanger) and Q-sepharose (anion exchanger) sorbents.

Recombinant mRBD and S-trimer proteins were isolated from the cultural medium of the CHO-K1 cells. The cultural medium was centrifuged to remove cell debris, filtered using −0.22 μm filters, and purified via subsequent Ni-NTA and ion-exchange chromatography as described above, skipping the refolding step.

Protein fractions were analyzed by SDS-PAGE in 15% separating polyacrylamide gel, and the target protein fraction was dialyzed against PBS.

The samples of the obtained proteins were sterilized by filtration through 0.22 µm filters. Gel-Pro Analyzer, Ver. 3.1 program determined purity and homogeneity. The quantitative analysis of the protein content was performed by the Lowry method.

### 2.4. Bio-Layer Interferometry

Recombinant human ACE2 protein (a.a. 18Q–740S, GenBank: AF291820) with C-terminal His-tag and Avi-tag was expressed in CHO-K1 cells. Purified protein was then chemically biotinylated with a fivefold molar excess of EZ-Link Sulfo-NHS-LC-Biotin (Thermo Scientific, USA) in PBS for 1 h at RT. Prepared ACE2 and the control irrelevant protein were diluted to 30 µg/mL in BLI buffer (PBS, 0.1% tween-20), and loaded onto streptavidin pins in an Octet K2 system (Pall ForteBio, Fremont, CA, USA). Subsequently, mRBD or pRBD were added at 30 μg/mL (285 nM) concentration to measure the association kinetics. The following 6-step scheme was used: baseline 60 s, loading 300 s, baseline2 30 s, association 180 s, dissociation 320 s, regeneration 6 × 5 s.

### 2.5. ELISA Assays

Serum samples of healthy donors were collected before the COVID-19 pandemic, and immune sera were obtained from patients confirmed with SARS-CoV-2 infections of varying severities. Blood samples were taken 2–3 weeks after symptom expression and confirmation of the diagnosis by PCR tests for SARS-CoV-2.

Serum samples were heated for 30 min at 56 °C and preincubated with 3 µg/mL of *E.coli* lysate for 1 h at 37 °C to adsorb anti-*E.coli* antibodies.

The pRBD, mRBD, and S-trimer proteins were coated to 96-well plates at 100 ng/well and incubated overnight. Plates were then washed with PBST buffer (0.1% Tween-20 in PBS) and blocked with 1% of casein in PBST for 1 h at RT. 100 µL of either human or mouse sera dilutions were added to the wells and incubated for 1 h at 37 °C. After washing three times with PBST, anti-human IgG HRP secondary antibodies (GenScript Piscataway, NJ, USA) or anti-mouse IgG HRP secondary antibodies (Sigma-Aldrich, St. Louis, MO, USA) were added and incubated for 1 h at 37 °C. The wells were washed again and TMB substrate solution (Amresco, Solon, OH, USA) was added to the wells. The reaction was then stopped with a 1N HCl solution, and absorbance was measured at 450 nm using Varioskan Lux multimode microplate reader (Thermo Fisher Scientific Inc., Waltham, MA, USA). Data were analyzed using GraphPad Prism 6.0 software.

### 2.6. Animal Immunization

All experimental protocols and procedures were approved by the SRC VB Vector Bioethics Committee (SRC VB Vector/10 September 2020 approved by the protocol of Bioethics Committee No. 5 as of 1 October 2020).

Groups of six female BALB/c mice were kept in separate cages under standard conditions, and had free access to food and water at all times.

Animals were immunized intraperitoneally twice in a two-week interval, with 80 µg of purified pRBD/mRBD resuspended in PBS or pRBD/mRBD in the presence of Incomplete Freund’s adjuvant (Sigma-Aldrich, Louis, MO, USA) or aluminum hydroxide (Brenntag Biosector A/S, Frederikssund, Denmark). The control group received intraperitoneal PBS injections. Blood samples were taken two weeks after the last immunization, incubated for 1 h at 37 °C and 2 h at 4 °C, and then centrifuged at 7000× *g* for 10 min. Sera were deactivated by heating for 30 min at 56 °C and stored at −20 °C.

### 2.7. Neutralization Assay for SARS-CoV-2

Neutralizing antibody titers were determined in the cytopathic effect (CPE) inhibition assays.

Vero E6 cells were seeded in 96-well plates and cultured for 24 h to form monolayers. Serial two-fold dilutions of serum samples were mixed at a 1:1 ratio with a solution of 100 TCID50 SARS-CoV-2 coronavirus strain nCoV/Victoria/1/2020 (obtained from State collection of causative agents of viral infections and rickettsioses SRC VB Vector, Russia) and incubated for 1 h at RT, and then added to a monolayer of Vero E6 cells. Plates were incubated for 4 days at 37 °C and were then stained with 0.2% gentian violet solution. The presence of specific CPE was assessed visually through the microscopic examination of the cell monolayer.

The dilutions of serum that completely prevented CPE in 50% of the wells were calculated by the Reed–Muench method [12].

### 2.8. Building of the RBD Model

Geometric parameters of the full-length SARS-CoV-2 S-protein (PDB:7BNN) were downloaded from the PDB database. Amino acid residues (319–560) corresponding to the RBD were isolated, and the missing hydrogen atoms were added. Unnecessary low-molecular-weight compounds were also removed, and the entire system was optimized in a limited force field OPLSe3 [13] for correct visualization of the secondary structure of the protein. The result was visualized by the VMD program [14].

## 3. Results

### 3.1. Construct Design, Expression, and Purification of pRBD, mRBD and S-Trimer

To provide the prokaryotic expression of RBD, we constructed a pET21-RBD expression vector harboring RBD gene fused with His-tag. *E.coli* BL21 (DE3) cells transformed with pET21-RBD plasmid were cultured in LB medium at 37 °C. Expression of the pRBD was induced with IPTG. After induction, the target protein was predominantly found in an inclusion body form. The purification of protein involved the sonication of bacterial cells, Ni-NTA affinity chromatography under denaturing conditions, the refolding of solubilized protein, and additional purification with ion exchange chromatography using Q- and SP-sepharose.

Using the bacterial expression system, we were able to obtain the ~27 kDa non-glycosylated pRBD protein (Figure 1A). The target protein yield was up to 90 mg/L.

For mammalian RBD expression, we constructed an integrative vector pVEAL2-RBD, providing the stable expression of the RBD domain fused with the Tpa signal sequence at the N-terminus and His-tag at the C-terminus of the protein. After the transfection of CHO-K1 cells, highly productive cell clones were selected and cultured for mRBD production. The cultural medium was clarified by centrifugation, mRBD was then purified using affinity and ion exchange chromatography.

Stably transfected CHO-K1 cells produced the ~35 kDa glycosylated mRBD (Figure 1), yielding up to 50–100 mg/L of the recombinant protein.

The purity of both pRBD and mRBD proteins exceeded 98%. Both proteins were dialyzed against PBS and sterilized by filtration.

Obtained SARS-CoV-2 S-trimer was used as a positive control for a comparative analysis of the antigenic properties of pRBD and mRBD since the trimerized RBD of the S protein most closely resembles coronaviral spike structures. The S-trimer was produced in CHO-K1 cells and purified as described above.

### 3.2. P. RBD and mRBD Characterization

The antigenicity of the recombinant pRBD, mRBD proteins, and S-trimer were assessed in ELISA using convalescent sera from donors who had previously positive RT-PCR tests for SARS-CoV-2 (Figure 1C).

All three proteins showed specific reactivity with COVID-19-positive sera, thus possessing antigenic properties similar to the corresponding SARS-CoV-2 viral proteins. However, *E.coli*-derived pRBD was less efficiently bound to specific serum IgGs from convalescent donors, compared with the mRBD and S trimer.

Next, we measured the interaction between both RBD variants and immobilized human ACE2 with biolayer interferometry (BLI). The equilibrium dissociation constant (KD) was 58.2 ± 1.3 nM using 285 nM mRBD as the soluble analyte. In contrast, no association with ACE2 was registered during the incubation with 285 nM pRBD.

### 3.3. PRBD and mRBD Immunogenicity in Mice

The immunogenicity of pRBD and mRBD was assessed in a BALB/c mice model. Incomplete Freund’s adjuvant (IFA) and aluminum hydroxide (Al(OH)_3_) were used as adjuvants.

Animals were immunized intraperitoneally twice on days 1 and 14, and blood sera were obtained from all mice two weeks after the booster immunization. The presence of RBD-specific IgGs in mice sera was detected by ELISA using pRBD, mRBD, and S-trimer as antigens (Figure 2A–C).

The data obtained indicate that the immunogenicity of mRBD is significantly higher than that of pRBD. As shown in Figure 2C, the S-trimer-specific antibody titer of sera from mice immunized with adjuvanted mRBD was ten-fold higher compared to sera from mice that received adjuvanted pRBD (about 1: 200,000 vs. 1: 20,000 on average). Similar ELISA results were obtained when pRBD and mRBD were used as antigens. However, less variation between anti-pRBD antibody titers in sera of mice from different groups was observed.

Thus, we found that after two immunizations, RBD obtained from mammalian cells and injected with adjuvant, elicited a much stronger specific humoral immune response in mice compared with the RBD of the prokaryotic origin, or mRBD without an adjuvant.

The neutralizing antibody titers of the animal immune sera were determined in the SARS-CoV-2 coronavirus strain nCoV/Victoria/1/2020 CPE inhibition assay in vitro.

It was shown that the RBD-specific antibody titers in mice immunized with mRBD were significantly higher and possessed about 100-fold higher neutralizing activity (Figure 2D).

Thus, mRBD induces a more appropriate humoral immune response in the BALB/c mice. It highlights the importance of correct folding and the glycosylation of proteins which can potentially be used as subunit vaccines.

## 4. Discussion

SARS-CoV-2 S glycoprotein mediates the binding of the virus to target cells through the ACE2 receptor (angiotensin-converting enzyme 2), ultimately leading to its penetration into cells [15,16] and the development of infection. S protein ectodomain has two main domains: the N-terminal domain (NTD) and the receptor-binding domain (RBD) [17,18,19,20]. It has been shown that RBD is highly immunogenic, and RBD-specific antibodies possess the neutralizing activity that prevents humans and animals from being infected [16,20,21]. An analysis of antibody epitopes that neutralize SARS-CoV-2 showed that such epitopes are mainly located in the RBD region [22,23,24,25,26,27,28]. Moreover, this domain is one of the most conserved regions of S glycoprotein [15,16,19,29]. That is why RBD is considered to be one of the main B-cell targets when developing SARS-CoV-2 vaccine [30,31,32,33,34,35,36]. Some researchers believe that the RBD-based vaccine will be safer than the full-length S protein vaccines [37].

When developing subunit vaccines, it is very important to choose an optimal expression system that provides the synthesis of the target protein while maintaining its antigenic and immunogenic properties [38]. Protein production in bacterial cells is a well-studied, accessible, rapid and inexpensive way to obtain the required amount of immunogen [38]. At the same time, in cases in which the correct folding of the protein of interest requires post-translational modifications (glycosylation, methylation, etc.), mammalian cells are usually used in order to obtain the target protein closest to its native state [1], which is important for the immunogenicity of the recombinant proteins. In the case of SARS-CoV-2, this is crucial, since S glycoprotein has up to twenty glycosylation sites, four of which are in the RBD domain (Figure 1B). Moreover, RBD is stabilized by disulfide bridges [39].

In this study, we engineered two expression constructs for the overproduction of the recombinant RBD protein, both in *E.coli* BL21 (DE3) and the CHO-K1 cell lines. The protein yield in the prokaryotic and mammalian expression systems was 90 and 50–100 mg/L of culture, respectively. The use of mammalian cells had several obvious advantages, primarily, correct folding and glycosylation by which the protein was secreted into the cultural medium. The process of protein purification from the cultural medium included standard methods of therapeutic protein chromatography. A major disadvantage of using a prokaryotic producer is the need to carry out the renaturation refolding procedure, since the proteins tend to accumulate in inclusion bodies. In addition, the *E. coli*-derived proteins are contaminated with endotoxins; the purification of such products is technologically difficult.

Using ELISA, we showed that both pRBD and mRBD proteins react with sera of donors who have recovered from COVID-19, and are not reactive towards healthy sera. However, pRBD had a lower affinity to convalescent sera, and BLI analysis revealed a poor affinity of pRBD to recombinant human ACE2 protein.

A mouse model analysis of the immunogenicity showed that both pRBD and mRBD elicit virus neutralizing cross-specific IgG antibodies. At the same time, the IgG titer determined by ELISA was much higher in the groups of animals immunized with the mRBD protein produced in mammalian cells.

The high specificity of sera taken from animals immunized with mRBD for SARS-CoV-2 S-protein trimers confirmed that mRBD has a large number of conformational epitopes (IEDB, https://www.iedb.org/home_v3.php (accessed on 11 December 2021), the formation of which is strongly influenced by post-translational modifications which are different in various expression systems. It is known that more than 50% of all human proteins and more than 40% of currently used pharmaceutical proteins are glycosylated. The biological activity of a protein, its pharmacodynamics, and immunogenicity depend on the correctness of glycosylation. The reason viruses are capable of efficiently evading the host’s immune system, protecting themselves from proteases, and interacting with cellular high affinity receptors, is that envelope proteins are glycosylated [40]. Therefore, the absence of some post-translational modifications of recombinant antigens can dramatically blunt the effectiveness of subunit vaccines.

Live virus neutralization assay of the animal immune serum showed that the protective humoral immune response of mRBD is much stronger than pRBD. Literature data, as well as the IEDB Database analysis (https://www.iedb.org/home_v3.php (accessed on 11 December 2021), show that the vast majority of neutralizing antibodies are formed against the conformational epitopes of RBD [41,42]. Probably, incorrect folding of pRBD leads to the loss of significant conformational epitopes and, consequently, to a decrease in the neutralizing activity of immunized animal sera.

The obtained results show that post-translational modifications provided by mammalian cells in the recombinant RBD protein are very important for its immunogenicity.

All of the aforementioned provides a basis to recommend the mammalian RBD protein that we have developed as a protective vaccine against COVID-19, inducing antibodies against the RBD domain of SARS-CoV-2.

## Figures and Tables

**Figure 1 vaccines-10-00096-f001:**
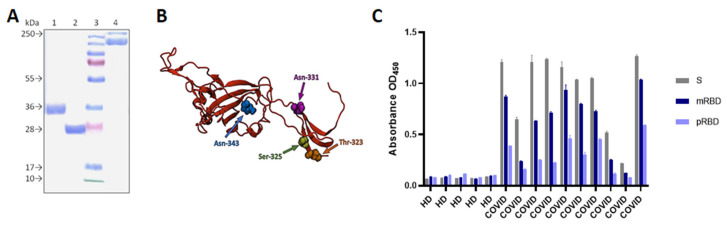
Recombinant proteins characterization. (**A**) SDS-PAGE analysis of purified recombinant SARS-CoV-2 proteins; 1—mRBD produced in CHO cells; 2—pRBD produced in *E.coli* cells; 3—molecular weight marker; 4—S-trimer produced in CHO cells. (**B**) RBD SARS-CoV-2 3D structure model visualized by VMD program. Glycosylated amino acid residues are indicated by arrows. (**C**) Evaluation of antigenicity of recombinant S-trimer, mRBD and pRBD proteins in ELISA using 10 COVID-19 convalescent sera (COVID) and 5 sera from healthy donors (HD) (dilution 1:100). Data represented as mean ±SD of three experiments.

**Figure 2 vaccines-10-00096-f002:**
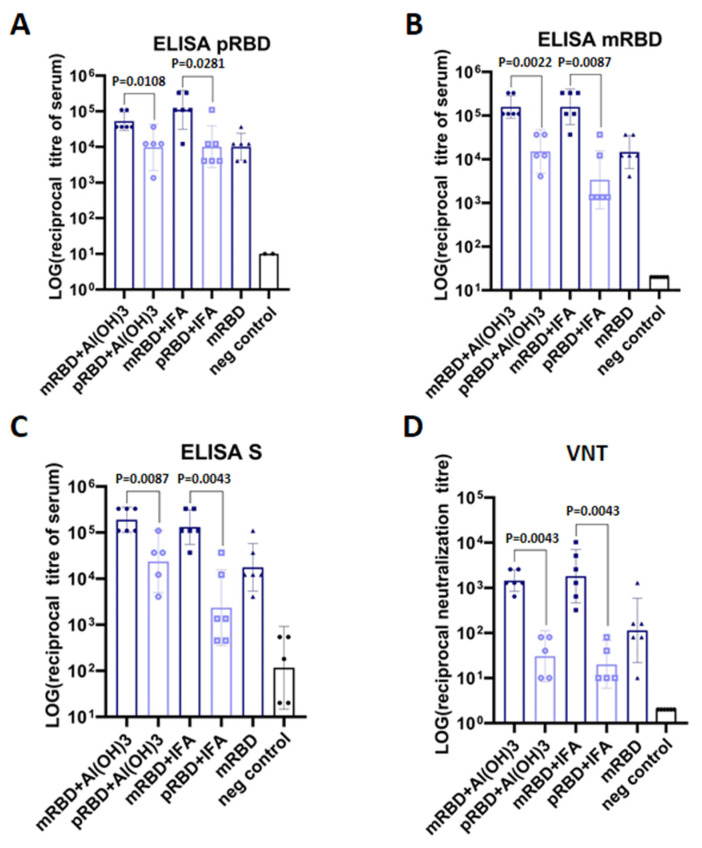
Immunogenicity of recombinant pRBD and mRBD in mice. Female BALB/c mice were immunized intraperitoneally days 0 and 14 with 80 µg pRBD/mRBD adjuvanted Al(OH)_3_ or Incomplete Freund’s adjuvant (IFA) or without adjuvant. The control group was immunized with PBS. Blood samples were collected on day 28 and tested for specificity to recombinant pRBD (**A**), mRBD (**B**), and S-trimer (**C**), and for neutralization activity against live SARS-CoV-2 virus (**D**). All the graphs and statistical analyses were performed using GraphPad Prism software. Data presented as geometric mean ±SD with the 95% confidence interval, statistical significance was calculated using nonparametric Mann–Whitney method.

## Data Availability

Not applicable.

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
