# Peer review of "Comparative Immunogenicity of the Recombinant Receptor-Binding Domain of Protein S SARS-CoV-2 Obtained in Prokaryotic and Mammalian Expression Systems"

_vaccines, 2022, doi:10.3390/vaccines10010096_

Round 1

Reviewer 1 Report

The authors compare SARS-CoV-2 spike protein receptor binding domain (RBD) expressed in E. coli (pRBD) or CHO cells (mRBD) for immunogenicity in mice and recognition by antibodies in human immune sera following COVID-19 infection. Comparison is made to recombinant trimeric full-length spike protein expressed in CHO cells as a gold standard. In all comparisons, mRBD was better than pRBD, consistent with the author’s speculation that glycosylation and post-translational folding in a eukaryotic expression system is functionally advantageous to a prokaryotic expression system. This finding is not surprising and is consistent with previous studies in other systems with other antigens, but it is an important and clear examination with great relevance for the current COVID-19 pandemic and development of effective vaccines against SARS-COV-2. The study was well designed, the results are clear and supporting of the authors’ interpretations and conclusions, and the manuscript is generally well written.

I have one specific comment and suggestion, which are minor. On line 30, replace “eukaryotic” with “mammalian.” The terms “pRBD” and “mRBD” are used frequently throughout the manuscript. The connection of “pRBD” with prokaryotic expression is clear at first use, on line 30. The meaning of “mRBD” to denote mammalian expression is much more diffuse and delayed in the current manuscript.

Reviewer 2 Report

It is an interesting paper,, however, I have some remarks which should be addressed by the Authors.:

  1. Abstract: far more it is quite a personal assessment it would be better to describe the difference more precisely
  2. Introduction: the information provided on the advantages and disadvantages of subunit vaccines should be supported by the appropriate references.
  3. Materials and methods section is nicely phrased and easy to follow, however, all crucial steps in the procedures described should be referred to procedures already employed for a similar purpose  to offer  a bit more universal message
  4. Time of blood collection should be more precisely provided using a time interval within which the samples were collected.
  5. Immunogenicity of the recombinant pRBD and mRBD in mice illustrated by the live virus neutralization assay suggests that pRBD immunization is even less effective in inducing neutralization antibodies than those identified in the ELISA.

I believe that the paper is worth to be published, however, I believe that being more competent than my reviewing the technical aspects, provided by other reviewers will give credit to the data presented.

Author Response

This manuscript is a resubmission of an earlier submission. The following is a list of the peer review reports and author responses from that submission.

Round 1

Reviewer 1 Report

The authors suggest a possibility of isolation of recombinant RBD from bacterial lysates as a low cost beneficial method of vaccine production, although their results show that the method is in fact not highly applicable, since immunogenicity of RBD isolated from bacteria is significantly lower than mammalian cell derived RBD. Since lower immunogenicity of bacterially derived RBD has been shown previously and the study does not provide improvements of already available techniques of recombinant RBD preparation, I do not see the novelity of the present study.

The authors fail to cite relevant literature on isolation of RBD from mammalian, yeast and bacterial cells in their introduction.

The authors state that 'The obtained results shows that post-translational modifications provided by mammalian cells in the recombinant RBD protein are very important for its immunogenicity.' I can not completely agree, that this is in fact what their results show. How can they be sure that lower immunogenicity is not due solely to improper protein folding and thus a lower concentration of properly folded protein in the pRBD sample? They show only degree of immunogenicity and do not explore the underlying cause for these differences.

Reviewer 2 Report

The authors presented a manuscript comparing production and immunogenicity of recombinant RBD made in either Ecoli or mammalian cells.

The idea is interesting as the need for recombinant protein is high, however, this manuscript suffers mainly from the following:

It is expected that the mammalian produced RBD is the preferred antigen for vaccine campaigns, however, the goal of this paper claims to be studying this in more detail. However, the authors do not properly quality control the produced proteins. If the bacterial RBD can be shown to be properly folded, non aggregated and still not be as immunogenic as the mammalian version, that would support the conclusions. At this point, no data shows this. Their low immunogenicity can clearly be just using low quality bacterial protein vs mammalian produced versions. 

1) The authors did not control any of their proteins. To link immunogenicity of the antigen the authors need to show their proteins are properly made and folded, not just a final gel. Standard things such as gel-filtration (at minimum), CD spectra, melting curves need to be shown.

2) Methods are incomplete. ie. refolding was just started as reduction in Urea with no other notes. 

Reviewer 3 Report

The manuscript entitled “Comparative immunogenicity of the recombinant receptor binding domain of protein S SARS-CoV-2 obtained in pro- and eukaryotic expression systems” covers proper methodology and discussed it very interestingly. Here, the authors have compared the humoral immune response elicited by the SARS-CoV-2 receptor-binding domain (RBD) produced using prokaryotic and eukaryotic expression systems. Based on their study, authors have reported that the RBD produced using eukaryotic expression system (CHO-K1 cells) elicits a greater immune response.  Another good point I noticed is that the authors have shown comparative ELISA titers for two different adjuvants. Overall, this manuscript is well written and includes sensible results and discussion. This manuscript significantly contributes to the ongoing COVID-19 vaccine research. This manuscript will be interesting to the readers of the Biomolecules.

I recommend authors work on the following minor comments to improve the quality of the manuscript.

Comment 1: Authors should include a size exclusion chromatography (SEC) profile of purified RBD constructs in Figure 1 as it gives an idea about the quality of purified constructs. The presence of soluble aggregates is of particular concern due to their negative effect on the efficacy and immunogenicity of immunogens which might be the case for E.coli produced RBD.

Comment 2: Authors need to correct the spelling: titre to “titer” (For example, it should be “Log (reciprocal titer of serum)”) Figure 3 (A), (B), (C), and (D).